# Steady and Fair Robustness Evaluation Based on Model Interpretation

## Abstract

Adversarial robustness has become a major concern as machine learning models are increasingly deployed in security-sensitive applications. Evaluating adversarial robustness remains a challenging task, as current metrics are heavily affected by various factors, including attack methods, attack intensities, and model architecture. In this paper, we propose Steady and Fair Robustness Evaluation, a novel framework designed to mitigate the impact of these factors and provide a more stable evaluation of a model's robustness. Our key insight is based on the strong correlation between the standard deviation (SD) of Shapley values, which measures the importance of individual neurons, and adversarial robustness. We demonstrate that models with lower SD of Shapley values are more robust to adversarial attacks, regardless of the attack method or model architecture. Extensive experiments across various models, training objectives, and attack scenarios show that our approach offers more consistent and interpretable robustness evaluation. We further introduce a new training strategy that incorporates the minimization of the SD of Shapley values for improving the robustness of the model. Our findings suggest that analysis based on Shapley value can provide a principled and efficient alternative to conventional robustness evaluation techniques.

## 1 Introduction

Adversarial robustness, the ability of machine learning models to resist adversarial attacks, has become increasingly crucial as deep learning is applied in security-sensitive domains. Numerous defense mechanisms have been developed to address adversarial vulnerabilities, yet the evaluation of these defenses remains limited in assessing accuracy (Wang et al., 2019; Rade & Moosavi-Dezfooli, 2022; Xu et al., 2023; Sehwag et al., 2020). While adversarial accuracy provides valuable insights when comparing models on the same attack method or the same architecture, it can introduce a bias towards the specific adversarial attacks used in the evaluation. To avoid this issue, a model needs to be tested against various adversarial attacks to evaluate its robustness. However, while this evaluation strategy that tests the adversarial accuracy of the network across multiple adversarial attacks can verify the robustness of the model, it sometimes leads to confusion.

Different adversarial attacks leverage different mechanisms to fool neural networks (Madry, 2018; Croce & Hein, 2020; Carlini & Wagner, 2017), which leads the models to show varying adversarial accuracy depending on the specific attack. This raises a fundamental question: *Which adversarial attacks should we trust for a fair evaluation?*

Many studies (Wang et al., 2019; Rade & Moosavi-Dezfooli, 2022; Xu et al., 2023; Sehwag et al., 2020) assess adversarial accuracy through extensive experiments using various attack strategies, which are complex and time-consuming. Also, Figure 1 demonstrates that evaluation relying on various attack strategies can lead to un-

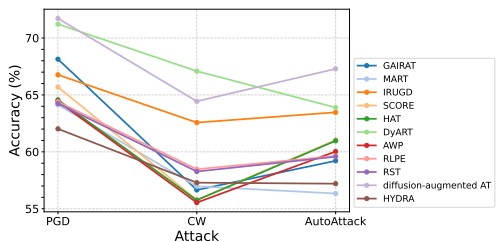

Figure 1: The performance of 11 defense strategies under three adversarial attack methods. It shows that the rank of models for adversarial robustness is highly dependent on the adversarial attack.

stable or sometimes inaccurate order for defense strategies. Figure 1 shows the performance of 11 different defense algorithms (Zhang et al., 2021a; Wang et al., 2019; Gowal et al., 2021; Pang et al., 2022; Rade & Moosavi-Dezfooli, 2022; Xu et al., 2023; Wu et al., 2020; Sridhar et al., 2022; Carmon et al., 2019; Wang et al., 2023; Sehwag et al., 2020) under PGD (Madry, 2018), CW (Carlini & Wagner, 2017), and AutoAttack (Croce & Hein, 2020). It shows that current adversarial accuracy highly depends on the types of adversarial attacks. On PGD and AutoAttack, Difusion-augmented AT (Wang et al., 2023) shows the highest performance among the compared methods. However, it shows vulnerability to CW attack, allowing it to be surpassed by DyART (Xu et al., 2023). On PGD and CW, DyART demonstrates a performance advantage over IRUGD (Gowal et al., 2021), with a gap exceeding 4%. However, when evaluated using AutoAttack, this difference shrinks to less than 0.5%. This discrepancy complicates the assessment of the models' adversarial robustness.

To address this challenge, we propose a Steady and Fair Robustness evaluation framework (SF Robustness) that does not rely on types of adversarial attacks or the architecture of the model. The fundamental idea of SF robustness is that the model is adversarially vulnerable if it heavily relies on a few predictive neurons for its decisions (i.e., the importance score of individual neurons is unevenly distributed such that only a small number of predictive neurons score high importance while the majority of neurons are regarded redundant), and the model is adversarially robust if it makes use of its internal neuron altogether (i.e., the importance score of internal neurons are evenly distributed). In this paper, we provide a theoretical demonstration of this relationship and experimental results supporting this tendency across a wide range of networks on different architectures on various datasets.

Based on this, we assess the robustness of models by examining the importance scores of internal neurons. Specifically, the reliance of the model on a few predictive neurons is reflected in the distribution of the importance score of each neuron. In this work, we adopt the Shapley value (Shapley, 1997; Kuhn & Tucker, 1953; Lundberg & Su-In, 2017; Sundararajan & Najmi, 2020) to measure the importance of internal neurons. Shapely value is a concept from Game Theory, which evaluates each property's individual and combining effects (Shapley, 1997; Kuhn & Tucker, 1953; Lundberg & Su-In, 2017; Sundararajan & Najmi, 2020). However, calculating the Shapley value in a neural network by definition is almost impossible due to the computational complexity. SF Robustness leverages Taylor approximation of the Shapley value introduced in Khakzar et al. (2021). By examining the distribution of the Shapley value, we can interpret the model's reliance on important neurons, which gives an important hint for evaluating the adversarial robustness of a neural network.

Section 2.1 provides a theoretical foundation for the correlation between adversarial robustness and Shapley value. In Section 2.2, we empirically demonstrate this correlation. Based on our analysis in Section 2, we introduce SF robustness in Section 3. In Section 4, we present extensive experiments to demonstrate a correlation between the SF Robustness and the performance of various defense strategies. Additionally, Section 5 discusses the relation between SF robustness and the performance of various models trained with data augmentation strategies under a one-step adversarial attack. In summary, our key contributions are:

- We establish a strong correlation between the standard deviation of Shapley values and adversarial robustness.

- We demonstrate that the standard deviation of Shapley values can be used as a proxy for evaluating adversarial robustness without relying on complex attacks. This gives an advantage for stable evaluation of adversarial robustness, which was varied by the attack selection.

- We introduce a novel add-on defense strategy that optimizes the standard deviation of Shapley values, achieving competitive performance with baseline defense methods.

## 2 ANALYSIS OF ADVERSARIAL ROBUSTNESS AND SHAPLEY VALUE

### 2.1 THEORATICAL ANALYSIS BETWEEN ROBUSTNESS AND SHAPLEY VALUE

Let $a_i^l$, $w_i^l$, $b_i^l$ denote the activation, weight, bias of a neuron $i$ in layer $l$, respectively. $f$ denotes the activation function. Then, the activation can be calculated as

$$a_i^l = f(z_i^l), \tag{1}$$

where, $z_i^l = w_i^l \cdot a_i^{l-1} + b_i^l$ denotes pre-activated feature of neuron $i$ in layer $l$. The change in the activation of a neuron to adversarial perturbation can be written as

$$\Delta a_i^l = a_i^l(x^{adv}) - a_i^l(x). \tag{2}$$

Adversarial training aims to reduce the Eq. 2 during training time. Since the adversarial perturbation $\delta$ is small, we can approximate the change in $a_i^l$ using a first-order Taylor expansion around $x$. The first-order approximation of the change in activation is given by

$$\Delta a_i^l \approx \frac{\partial a_i^l}{\partial x} \cdot \delta. \tag{3}$$

Let $L$ as the final layer of the model (i.e., classification layer), first-order Taylor approximation of the penultimate layer Shapley value $s_i^{L-1}$ can be written as

$$S_i^{L-1} \approx a_i^{L-1} \nabla_{a_i^{L-1}} f(Z_i^l). \tag{4}$$

From Eq. 1, Eq. 4 can be written as follows:

$$S_i^{L-1} \approx a_i^{L-1} \nabla_{a_i^{L-1}} a_i^L, \tag{5}$$

$$S_i^{L-1} \approx a_i^{L-1} \cdot \frac{\partial a_i^L}{\partial a_i^{L-1}}. \tag{6}$$

From Eq. 3, activation difference in last layer $L$ is:

$$\frac{\partial a_i^L}{\partial x} \approx \Delta a_i^L \cdot \frac{1}{\delta}. \tag{7}$$

By using the chain rule,

$$\frac{\partial a_i^L}{\partial x} = \frac{\partial a_i^L}{\partial a_i^{L-1}} \cdot \frac{\partial a_i^{L-1}}{\partial x}. \tag{8}$$

From Eq. 6,

$$\frac{\partial a_i^L}{\partial a_i^{L-1}} \approx \frac{S_i^{L-1}}{a_i^{L-1}}. \tag{9}$$

Then, Eq. 8 can be written as follows:

$$\Delta a_i^L \cdot \approx S_i^{L-1} \cdot \frac{\delta}{a_i^{L-1}} \cdot \frac{\partial a_i^{L-1}}{\partial x}. \tag{10}$$

It can be interpreted that neurons with higher Shapley values experience larger changes in activation due to adversarial perturbations.

Eq. 10 indicates that minimizing $S_i^{L-1}$ can achieve less activation difference, referring to more adversarial robustness. However, the total Shapley value of a layer cannot be zero due to the nature of the Shapley value. Zero Shapely value means zero contribution to the output, which cannot happen in the neural network's layer unless it returns the same value to the next layer. Consequently, in the robust model, the importance of individual neurons should be more evenly distributed, with Shapley values clustering closer to zero. This reduces the model's reliance on any single neuron, making it more robust to adversarial attacks.

## 2.2 EMPIRICAL ANALYSIS BETWEEN ROBUSTNESS AND SHAPLEY VALUE

In standard training (i.e., training with non-adversarial inputs), models often rely on specific neurons that are highly predictive (Ilyas et al., 2019). Also, recent studies showed robust and non-robust features in the model contribute to the model's robustness (Ilyas et al., 2019; Kim et al., 2021). White-box attacks exploit this reliance by targeting these non-robust feature neurons to compromise the model's performance.

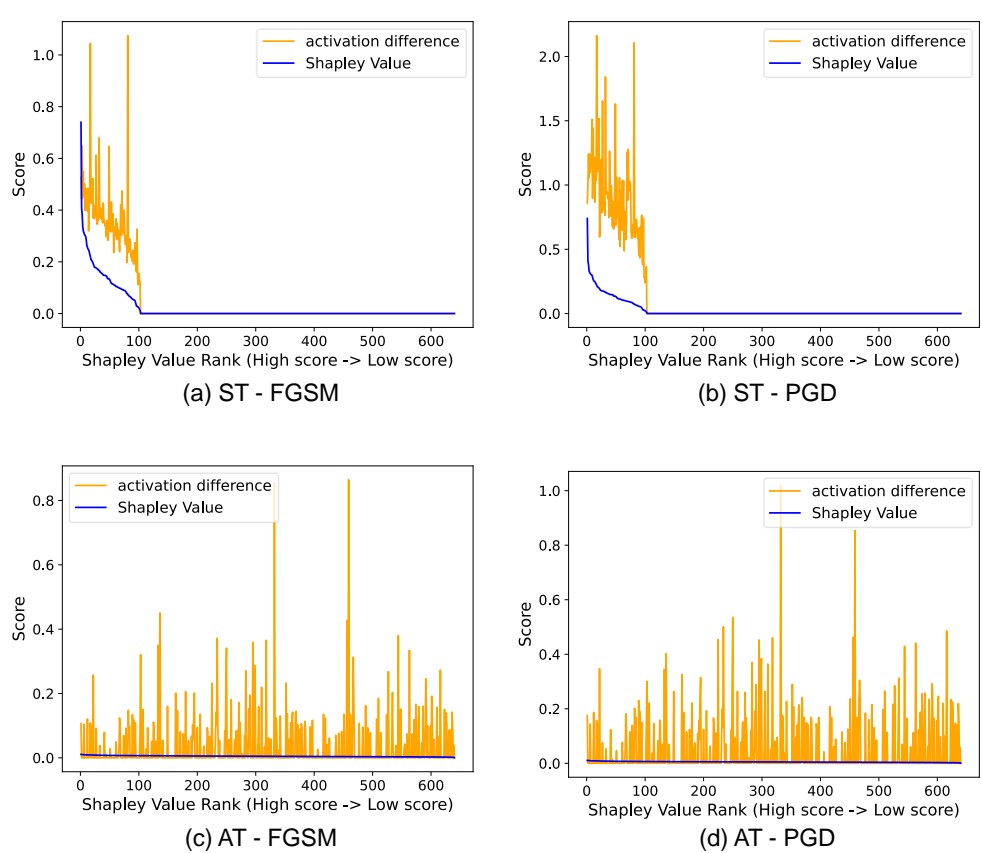

Figure 2: Comparison between the standard trained (ST) model and adversarial trained (AT) model (Wang et al., 2019). We computed the activation difference between the clean and adversarial samples generated by FGSM and PGD. The neurons are sorted based on Shapley value in descending order. Important neurons (i.e., neurons with high Shapley value) in ST models are much more targeted for adversarial attacks than those of AT models.

Accordingly, we analyze what happens in standard training. In a standard trained model, some neurons (i.e., non-robust neurons) might respond extremely to adversarial perturbations, causing $a_i^l(x^{adv})$ to be far from $a_i^l(x)$. On the other hand, other neurons (i.e., robust neurons) might respond weakly or be unaffected by adversarial perturbation.

Figure 2 illustrates the activation differences in the penultimate layer of standard-trained models and adversarially trained models, denoted by ST and AT, respectively. In standard trained models, neurons with high importance (i.e., neurons with high Shapley values) show larger activation differences under adversarial attack. The result matches the correlation between the Shapley value and activation difference in Eq. 10, also suggesting that white-box adversarial attacks target important neurons more. This results in large variations in neuron activations across entire neurons because some neurons exhibit significant changes in activation while others show little to no change. This variability contributes to a higher standard deviation of Shaple value across neurons.

In contrast, neurons of adversarially trained models show smaller changes in activation under adversarial attack, which leads to more consistent activations across neurons when comparing the clean and adversarial inputs. In other words, the activations across neurons become more uniform, which directly reduces the standard deviation of the neuron activations across $i$. This means that in adversarial training, the model shifts away from over-relying on a few important neurons and instead distributes its focus across a broader set of neurons that are more stable under adversarial perturbations. Consequently, the importance of individual neurons in adversarially trained models is more

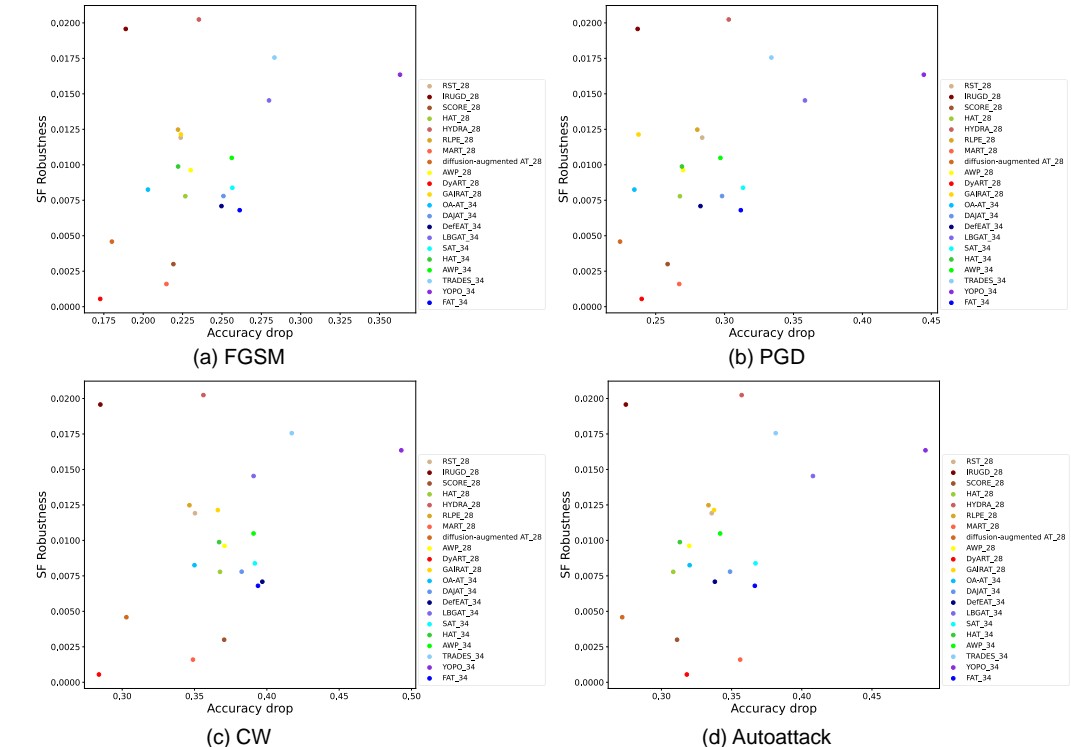

Figure 3: SF robustness and accuracy drop of various defense methods. We display two different architectures, WRN28-10 and WRN34-10, in the CIFAR-10 dataset with 21 different defense strategies. Accuracy drop computed on four different attacks (FGSM (Goodfellow et al., 2015), CW (Carlini & Wagner, 2017), PGD (Madry, 2018), and AutoAttack (Croce & Hein, 2020)). _28 and _34 refers model architecture, WRN28-10 and WRN34-10, respectively.

evenly distributed, with Shapley values clustering closer to zero. This balance reduces the model's reliance on any single neuron, making it more robust to adversarial attacks.

## 3 SF ROBUSTNESS

Let target model $g$, input $x \in D_{\text{train}}$, the number of images in $D_{\text{train}}$ as $n$, Taylor approximation of Shapley value for $i$-th neuron in the penultimate layer $c_i^L(x)$ is calculated as

$$c_i^{L-1}(x) = a_i^{L-1} \nabla_{a_i^{L-1}} g(x). \tag{11}$$

However, $c_i^{L-1}(x)$ can only capture the Shapley value of a single input. We calculate the average of the Shapley value over the training set to interpret the global response of the target model. A global Shapley value $C_i^{L-1}$ can be defined as

$$C_i^{L-1} = \frac{1}{n} \sum_{x \in D_{\text{train}}} (c_i^{L-1}(x)). \tag{12}$$

Note that the global Shapley value is calculated only with the clean sample. From the calculated global Shapley value $C_i^L$, SF Robustness of a model $g$ is calculated as follows:

$$\text{SF}(g) = \sqrt{\frac{\sum_{i=1}^{k} (C_i^{L-1} - m)^2}{k}}, \tag{13}$$

where $k$ denotes the number of neurons in a penultimate layer of the target model, and $m$ denotes the mean of the global Shapley values $C^{L-1} = \{C_1^{L-1}, C_2^{L-1}, \cdots, C_k^{L-1}\}$.

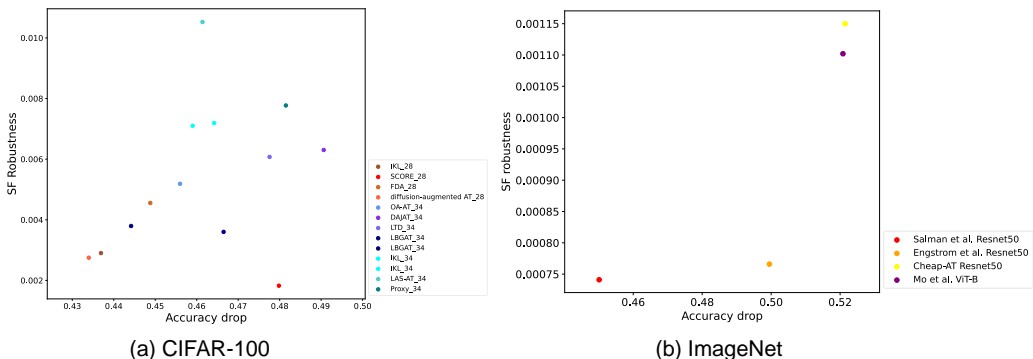

(a) CIFAR-100                    (b) ImageNet

Figure 4: SF Robustness and accuracy drop calculated from various defense methods. In (a), we display two different architectures, WRN28-10 and WRN34-10, in the CIFAR-100 dataset with 13 different defense strategies. _28 and _34 refers model architecture, WRN28-10 and WRN34-10, respectively. In (b), we show ResNet50 and ViT-B in the ImageNet dataset, with four different defense strategies. Accuracy drop is computed as an average drop from four different attacks (FGSM (Goodfellow et al., 2015), CW (Carlini & Wagner, 2017), PGD (Madry, 2018), and AutoAttack (Croce & Hein, 2020)).

## 4 EXPERIMENTS

In this section, we assess various adversarially trained models by evaluating their SF Robustness and adversarial accuracy on CIFAR-10, CIFAR-100, and ImageNet. We define accuracy drop $\Delta Acc = 1 - \frac{adv\_acc}{clean\_acc}$ as the ratio of the adversarial performance ($adv\_acc$) compared to the clean sample performance ($clean\_acc$). Accuracy drop assesses the amount of performance drop due to the adversarial attack by considering the original clean sample performance.

For this study, we leverage a total of 38 available pre-trained weights for WRN28-10 and WRN34-10 trained on CIFAR-10, CIFAR-100, and ResNet50, ViT-B trained on ImageNet released by Croce et al. (2021). Detailed information on pre-trained models is provided in Appendix A.3.1. Throughout the experiments, we have found a strong correlation between SF Robustness and the adversarial robustness of the model. This indicates SF robustness can serve as a powerful metric for evaluating adversarial robustness within different architectures and defense strategies.

### 4.1 MODEL COMPARISON BASED ON SF ROBUSTNESS

#### 4.1.1 EVALUATION ON CIFAR BENCHMARK

**CIFAR-10.** In Figure 3, we evaluated the effectiveness of SF Robustness regarding the four different attacks: FGSM, PGD, CW, and AutoAttack. Enlarged figures are provided in Appendix A.3.2. On CIFAR-10, we used two model architectures (WRN28-10 and WRN34-10) with 21 different defense strategies. Detailed information on pre-trained weights is provided in Appendix A.3.1. The X-axis of the figure illustrates the accuracy drop rate. Lower values on this axis correspond to models with higher robustness. The Y-axis of the figure is SF Robustness (i.e., the standard deviation of the Shapley values), and the lower value indicates a smaller standard deviation (SD) of the Shapley values. Red series colors indicate WRN28-10 models and blue series colors indicate WRN34-10 models. Experimental results showed that models with low SF robustness tend to score high adversarial robustness on all four attack methods. Despite figure 3 displaying different architectures in one figure, it still shows a strong correlation between the SF Robustness and the adversarial robustness of the model.

Overall, we can observe a positive correlation between SF robustness and adversarial robustness (i.e., a model with a low SF robustness tends to be more robust to adversarial attacks). This tendency suggests that SF robustness can function as a proxy metric to evaluate the adversarial robustness of the networks without relying on the choice of adversarial attack.

Table 1: Base loss for PGDAT and MART. CE and BCE denote cross-entropy and boosted cross-entropy loss, respectively. $x$ is clean example, $x'$ is adversarial example, and $y$ is the label. KL represents Kullback-Leibler divergence.

| Defense Method | Base Loss |
|---|---|
| PGDAT | $\text{CE}(g(x'), y)$ |
| MART | $\text{BCE}(g(x'), y) + \lambda_m \cdot \text{KL}(g(x) \parallel g(x')) \cdot (1 - g_y(x))$ |

Table 2: Comparison between model training with and without standard deviation(SD) of Shapley value as regularization. $\downarrow$ denotes smaller is better.

| Method | SD $\downarrow$ | PGD | CW | AutoAttack |
|---|---|---|---|---|
| PGDAT | 0.01476 | 54.25 | 48.20 | 49.82 |
| PGDAT + SF (Ours) | **0.01297** | **55.18** | **49.34** | **50.73** |
| MART | 0.01293 | 56.17 | 49.45 | 50.92 |
| MART + SF (Ours) | **0.01153** | **56.78** | **49.95** | **51.07** |

**CIFAR-100.** Evaluation on CIFAR-100 tells a similar story. Figure 4 (a) shows SF robustness and corresponding adversarial accuracy averaged over four adversarial attacks (FGSM, PGD, CW, and AutoAttack). Experimental results on individual attack strategies are given in Appendix A.3.3. We used two model architectures (WRN28-10 and WRN34-10) with 13 different defense strategies. Detailed information on pre-trained weights is provided in Appendix A.3.1. Experimental results show that SF robustness and adversarial robustness are closely related.

### 4.1.2 EVALUATION ON IMAGENET BENCHMARK

In Figure 4 (b), we verified SF Robustness on ImageNet under four adversarial attacks (FGSM, PGD, CW, and AutoAttack). Results of each adversarial attack method are provided in Appendix A.3.4. We used ResNet50 and ViT-B with four different defense strategies on the ImageNet dataset. On ImageNet, we found a similar tendency observed in Section 4.1.1. Detailed information on pre-trained weights is provided in Appendix A.3.1.

### 4.2 CAN SF ROBUSTNESS MAKE THE MODEL MORE ROBUST?

So far, we have demonstrated the positive correlation between SF robustness and adversarial robustness of the model. Then, a natural question arises: can we build an adversarially robust network by minimizing SF robustness of the network?

In this subsection, we use SF robustness as an additional defense strategy and show that minimizing the SD of the Shapley value leads to robustness improvement. We define SF robustness constraints as below

$$\mathcal{L}_s = \text{SF}(g), \tag{14}$$

where $\text{SF}(g)$ is determined by Eq. 13 where global Shapley value $C_i^{L-1}$ are calculated by Eq. 12 with respect to $D_{\text{minibatch}}$. Then, the overall training objective can be formulated by

$$\mathcal{L} = \mathcal{L}_{base} + \lambda \cdot \mathcal{L}_s, \tag{15}$$

where $\mathcal{L}_{base}$ is the base loss. $\lambda$ is a hyperparameter to balance the base loss and SF robustness constraints. In this work, we verify the effectiveness of SF robustness as a training objective with three different adversarial training base losses defined by PGDAT (Madry, 2018) and MART (Wang et al., 2019) and the base loss for PGDAT and MART is summarized in Table 1.

We train 20 additional epochs with SF robustness on the adversarially pre-trained WRN28-10 on CIFAR-10. $\lambda$ is set to 1. Detailed training conditions are provided in Appendix A.1. Table 2 shows that SF robustness introduces adversarial robustness of the network on PGD, CW, and AutoAttack.

## 5 DISCUSSION

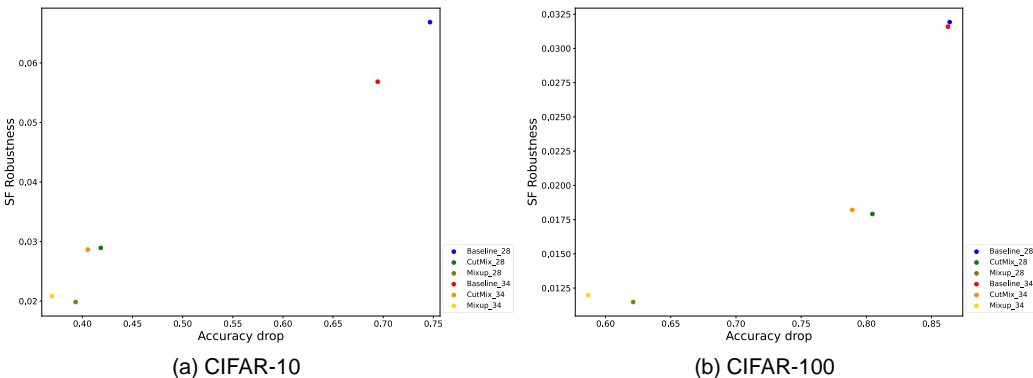

(a) CIFAR-10               (b) CIFAR-100

Figure 5: Comparison of models trained with various data augmentation strategies. We compared two different architectures, WRN28-10 and WRN34-10, in (a) CIFAR-10 and (b) CIFAR-100 datasets with three different data augmentation strategies. There is a positive correlation between SF Robustness and accuracy drop rate under FGSM (Goodfellow et al., 2015).

**SF Robustness and the performance of models trained with data augmentation.** Data augmentations such as CutMix and Mixup are known to boost the adversarial robustness of a model when applied with adversarial training (Rebuffi et al., 2021). Also, they have shown effectiveness in performance improvement on single-step adversarial robustness such as FGSM on standard training (Zhang et al., 2021b; Yun et al., 2019; Lamb et al., 2019). In this section, we show that the performance gain of these augmentation strategies under FGSM attack can be explained by SF robustness.

We assess WRN28-10 and WRN34-10 trained on CIFAR-10 and CIFAR-100 with CutMix (Yun et al., 2019), Mixup (Zhang, 2018) and baseline augmentation strategies. Detailed training setting is provided in the Appendix A.2. Figure 5 shows the SF robustness and performance under FGSM attack including Baseline (a model trained with baseline data augmentation such as resizing, cropping, and flipping), CutMix, and Mixup. As with adversar-

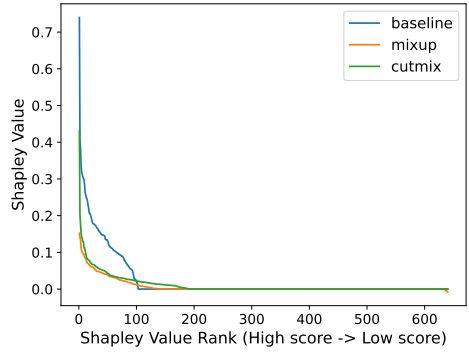

Figure 6: Distribution of Shapley value of models trained with baseline augmentation, Mixup, and CutMix. The neurons are sorted based on Shapley value in descending order. Mixup, which exhibits the best performance under FGSM and best SF robustness, shows the smoothed distribution of Shapley values.

ially trained models, we observe a high correlation between SF robustness and the accuracy drop rate. The baseline WRN28-10 trained with baseline data augmentation showed the highest shapley value among the compared models and the biggest accuracy drop rate. CutMix outperforms baseline on both SF robustness and accuracy drop under FGSM. The model trained with Mixup shows the best performance under FGSM than its counterparts, which is also reflected in the smallest SF robustness.

The performance improvement under FGSM on models trained with CutMix and Mixup can be explained by the smoothed distribution (i.e., small SD) of the Shapley value of internal neurons. Figure 6 shows the distribution of Shapley values of WRN28-10 trained with baseline augmentation, Mixup, and CutMix on CIFAR-10. Mixup, which performs best under FSGM shows the most smoothed distribution of Shapley values, resulting in the smallest SF robustness. On the other hand, the vulnerability of the baseline under FGSM can be explained by its distribution of Shapley values with high standard deviation.

Furthermore, we observe that WRN28-10 and WRN34-10 show similar performance under FGSM when trained with the same training strategies, and this tendency is also reflected in a small gap of SF robustness within the same training methods on different architecture. We also present our analysis on ImageNet in Appendix A.3.5.

# 6 RELATED WORKS

## 6.1 NEURON-WISE ANALYSIS ON ADVERSARIAL EXAMPLES

Kim et al. (2021) divided features (neurons) into two groups: robust features and non-robust features. They utilized an information bottleneck for the feature distillation. The noise is inserted in the intermediate layer to control the information flow. The noise is optimized in a way that minimizes the mutual information between the input and layer-inserted intermediate layer and maximizes the information between the prediction and the ground truth. This is in line with our work as they analyze the robustness of individual neurons. However, their approach is unsuitable for a model assessment because the noise is optimized to output the ground truth label. Therefore, it is not appropriate to quantify the properties of the network itself.

Zhang et al. (2020a) defined *neuron sensitivity* as the activation difference on adversarial and clean samples. They found that sensitive neurons (i.e., neurons that undergo significant changes on adversarial attack) play important roles in causing misclassification. This work is in line with our work in that their definition of sensitive neurons is associated with important neurons. However, neuron sensitivity requires adversarial attacks, and the extent to which each attack affects neuron activation is different. On the other hand, SF robustness does not rely on the types of adversarial attacks, which makes the evaluation process simple and stable.

## 6.2 EVALUATION OF ADVERSARIAL ROBUSTNESS

There are several adversarial robustness metrics other than adversarial robustness. **Minimal Perturbation** is the smallest perturbation added to and input that changes the model prediction which is also utilized in Carlini & Wagner (2017). **Probabilistic Accuracy** (Robey et al., 2022) is calculated by the proportion of correctly classified adversarial examples where the predicted probability is above a certain tolerance level. **Robustness w.r.t. predictions** (Ding et al., 2019) calculates accuracy on adversarial examples in which the perturbations are produced to perturb the model's original prediction on the clean examples instead of the true label. While they provide more fine-grained measurement than adversarial accuracy, they still highly depend on adversarial attacks and do not provide neuron-level inspection. **Adversarial Sparsity** (Olivier & Raj, 2023) shares a similar concept with local intrinsic dimensionality (Ma et al., 2018) where they characterize the dimensional properties of adversarial subspaces. Adversarial Sparsity considers the number of adversarial regions around the sample to quantify adversarial robustness. Their approach is based on the latent space, whereas our metric focuses on the internal responses of individual neurons.

# 7 CONCLUSION

In this paper, we introduced the Steady and Fair Robustness Evaluation (SF Robustness) framework, which addresses the challenges of inconsistencies in adversarial robustness evaluation caused by varying hyperparameters, model architectures, and attack methods. By leveraging the standard deviation of Shapley values as a key metric, we demonstrated a strong correlation between neuron importance variability and adversarial robustness. This finding allows for a more principled and interpretable evaluation of robustness, independent of attack specifics or model configurations. Through extensive experimentation, we showed that SF Robustness provides a reliable indicator of adversarial robustness, with models exhibiting lower SF Robustness demonstrating stronger resistance to attacks. Furthermore, we proposed a novel attack and defense strategy that optimizes the SD of Shapley values, outperforming the baseline defenses. Our results validate that SF Robustness can serve as an effective tool for both evaluating and improving adversarial robustness. This work opens the door for further exploration of interpretability-driven robustness strategies, ultimately contributing to more secure and reliable machine learning systems in adversarial settings.

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

# A    APPENDIX

## A.1    TRAINING DETAILS FOR ADVERSARIAL TRAINING

In experiments in Section 4.2, PGDAT is pre-trained with batch size 128 for 150 epochs with an initial learning rate 0.1. Pre-trained MART is trained with batch size 128 for 90 epochs with an initial learning rate of 0.1 decayed by 0.0035 at every 30 epochs.

On top of this, we run 20 additional epochs with SD of Shapley value regularization. Batch size, learning rate, and $\lambda$ are set to 128, 1e-5, and 1, respectively.

## A.2    TRAINING DETAILS FOR STANDARD TRAINING

We trained WRN28-10 and WRN34-10 on CIFAR-10 and CIFAR-100 with batch size 64 for 300 epochs. The initial learning rate is set to 0.25 and decayed by 0.1 at 150 and 225 epochs.

We use pre-trained ResNet-50 model weights released by Yun et al. (2019). The models are trained with batch size 256 for thirty epochs. The initial learning rate is 0.1 and decayed by a factor of 0.1 at 75, 150, and 225 epochs. Models trained with Mixup and CutMix are trained along with standard data augmentation strategies such as flipping, cropping, and resizing.

## A.3    DETAILED EXPERIMENT RESULTS

### A.3.1    PRETRAINED WEIGHTS.

For this study, we leverage a total of 38 available pre-trained weights for WRN28-10 and WRN34-10 trained on CIFAR-10, CIFAR-100, and ResNet50, ViT-B trained on ImageNet released by Croce et al. (2021). The works are listed below:

**WRN28-10, CIFAR-10.** DyART (Xu et al., 2023), MART (Wang et al., 2019), SCORE (Pang et al., 2022), diffusion-augmented AT (Wang et al., 2023), HAT (Rade & Moosavi-Dezfooli, 2022), AWP (Wu et al., 2020), RST (Carmon et al., 2019), GAIRAT (Zhang et al., 2021a), RLPE (Sridhar et al., 2022), IRUGD (Gowal et al., 2021), HYDRA (Sehwag et al., 2020)

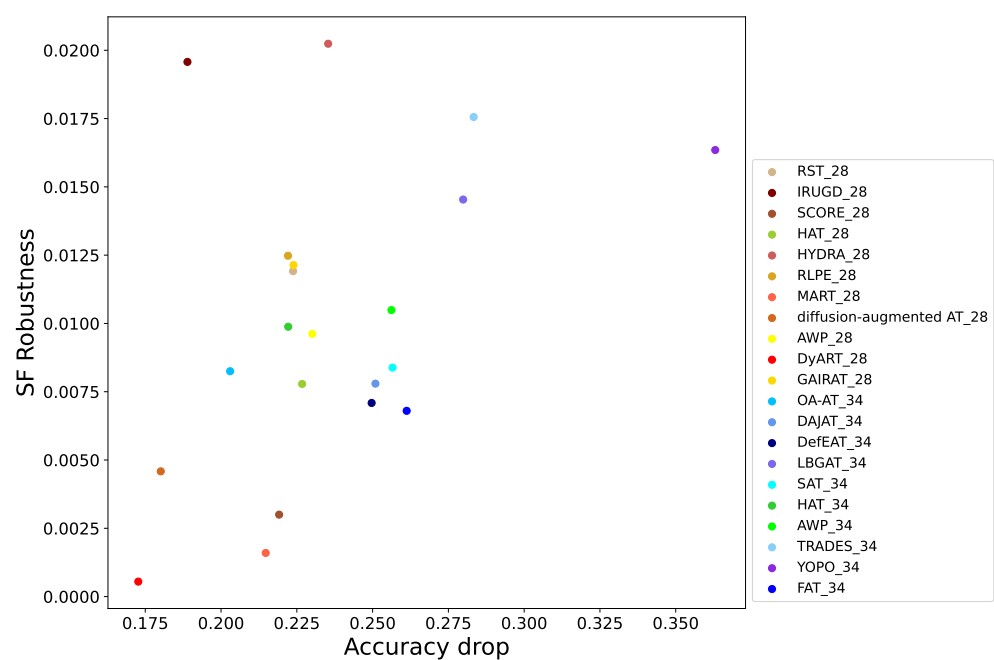

Figure 7: SF robustness and accuracy drop of various defense methods on FGSM in CIFAR-10. We compared two different architectures, WRN28-10 and WRN34-10, in the CIFAR-10 with 21 different defense strategies. Accuracy drop computed on FGSM (Goodfellow et al., 2015). _28 and _34 refers model architecture, WRN28-10 and WRN34-10, respectively.

**WRN28-10, CIFAR-100.** SCORE (Pang et al., 2022), diffusion-augmented AT (Wang et al., 2023), IKL (Cui et al., 2024), FDA (Rebuffi et al., 2021)

**WRN34-10, CIFAR-10.** FAT (Zhang et al., 2020b), DefEAT (Chen & Lee, 2024), DAJAT (Addepalli et al., 2022b), OA-AT (Addepalli et al., 2022a), SAT (Huang et al., 2020), HAT (Rade & Moosavi-Dezfooli, 2022), AWP (Wu et al., 2020), LBGAT (Cui et al., 2021), YOPO (Zhang et al., 2019a), TRADES (Zhang et al., 2019b)

**WRN34-10, CIFAR-100.** LBGAT (Cui et al., 2021), OA-AT (Addepalli et al., 2022a), IKL (Cui et al., 2024), Proxy (Sehwag et al., 2022), LAS-AT (Jia et al., 2022)

**RN50, Imagenet.** Salman et al. (Salman et al., 2020), Engstrom et al. (Engstrom et al., 2019), Cheap-AT (Wong et al., 2020)

**Transformers, Imagenet.** (ViT-B) Mo et al. (Mo et al., 2022)

**RN50, Imagenet, Discussion** Baseline (He et al., 2016), Mixup (Zhang, 2018), Cutmix (Yun et al., 2019)

A.3.2    Evaluation on CIFAR-10 Benchmark

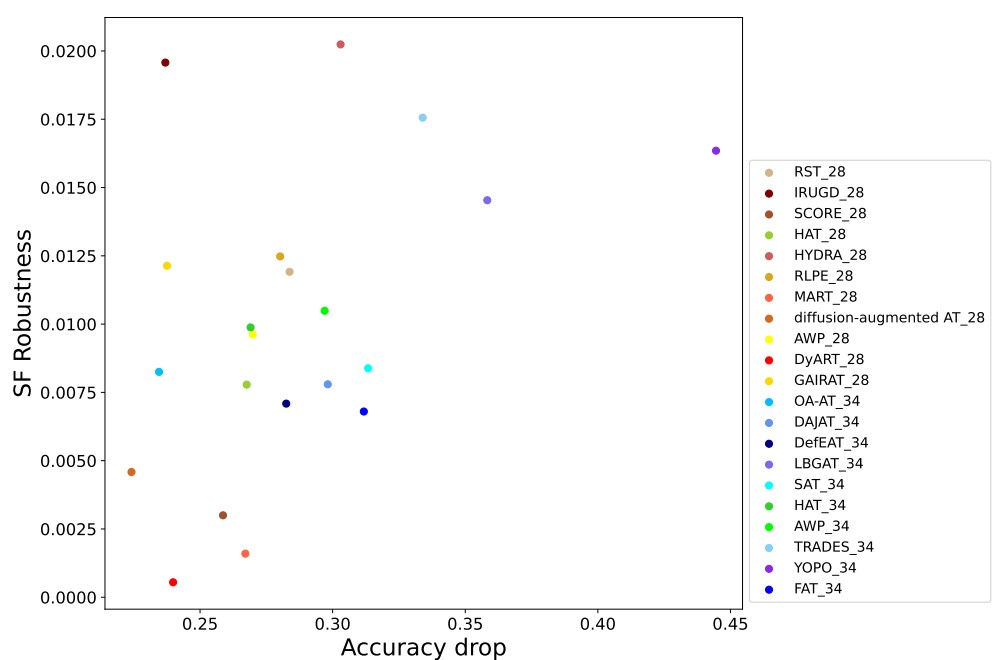

Figure 8: SF robustness and accuracy drop of various defense methods on PGD in CIFAR-10. We compared two different architectures, WRN28-10 and WRN34-10, in the CIFAR-10 with 21 different defense strategies. Accuracy drop computed on PGD (Madry, 2018). _28 and _34 refers model architecture, WRN28-10 and WRN34-10, respectively.

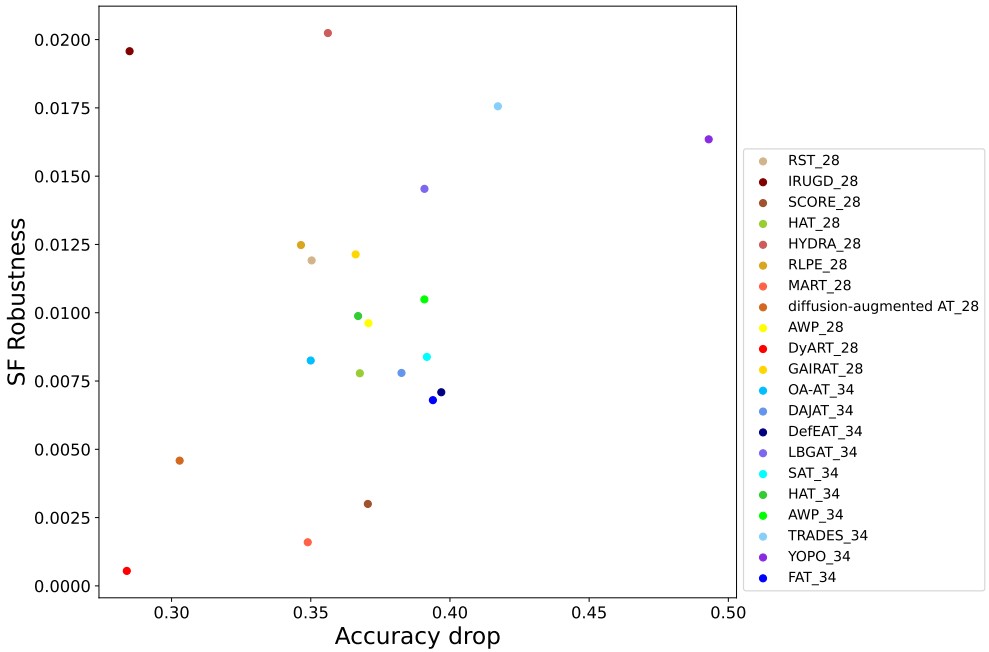

Figure 9: SF robustness and accuracy drop of various defense methods on CW in CIFAR-10. We compared two different architectures, WRN28-10 and WRN34-10, in the CIFAR-10 with 21 different defense strategies. Accuracy drop computed on CW (Carlini & Wagner, 2017). _28 and _34 refers model architecture, WRN28-10 and WRN34-10, respectively.

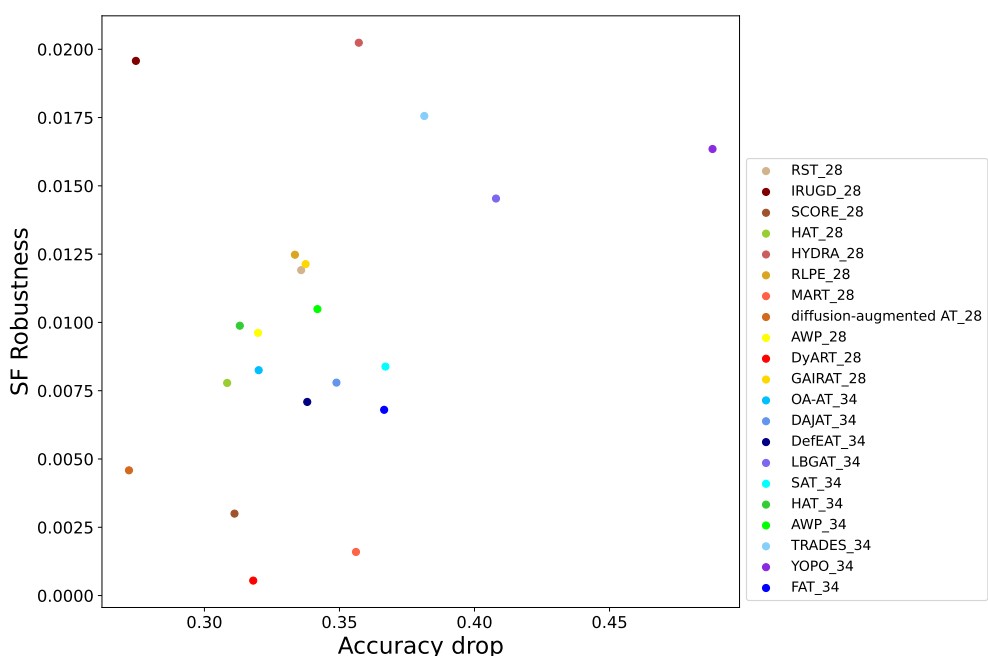

Figure 10: SF robustness and accuracy drop of various defense methods on Autoattack in CIFAR-10. We compared two different architectures, WRN28-10 and WRN34-10, in the CIFAR-10 dataset with 21 different defense strategies. Accuracy drop computed on Autoattack (Croce & Hein, 2020). _28 and _34 refers model architecture, WRN28-10 and WRN34-10, respectively.

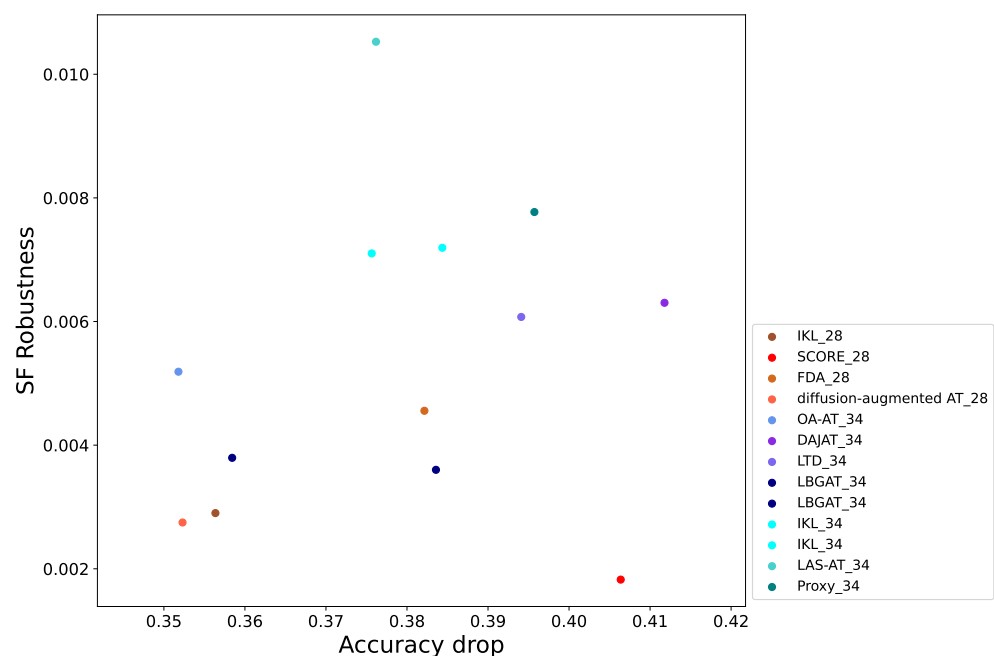

Figure 11: SF robustness and accuracy drop of various defense methods on FGSM in CIFAR-100. We compared two different architectures, WRN28-10 and WRN34-10, in the CIFAR-100 with 13 different defense strategies. Accuracy drop computed on FGSM (Goodfellow et al., 2015), CW (Carlini & Wagner, 2017), PGD (Madry, 2018), and Autoattack (Croce & Hein, 2020)). _28 and _34 refers model architecture, WRN28-10 and WRN34-10, respectively.

### A.3.3    EVALUATION ON CIFAR-100 BENCHMARK

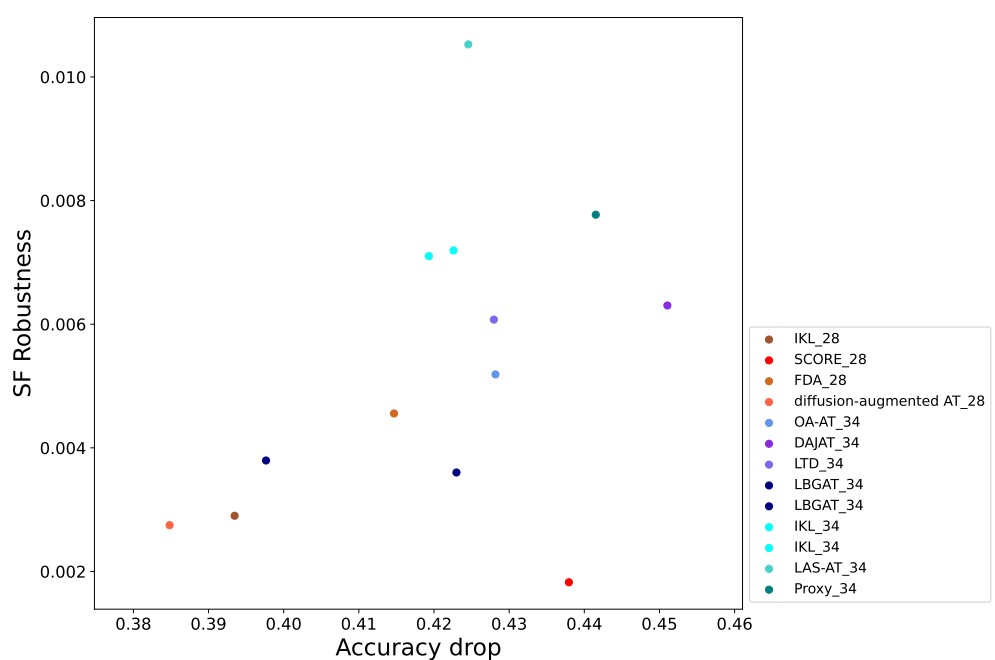

Figure 12: SF robustness and accuracy drop of various defense methods on PGD in CIFAR-100. We compared two different architectures, WRN28-10 and WRN34-10, in the CIFAR-100 with 13 different defense strategies. Accuracy drop computed on PGD (Madry, 2018). _28 and _34 refers model architecture, WRN28-10 and WRN34-10, respectively.

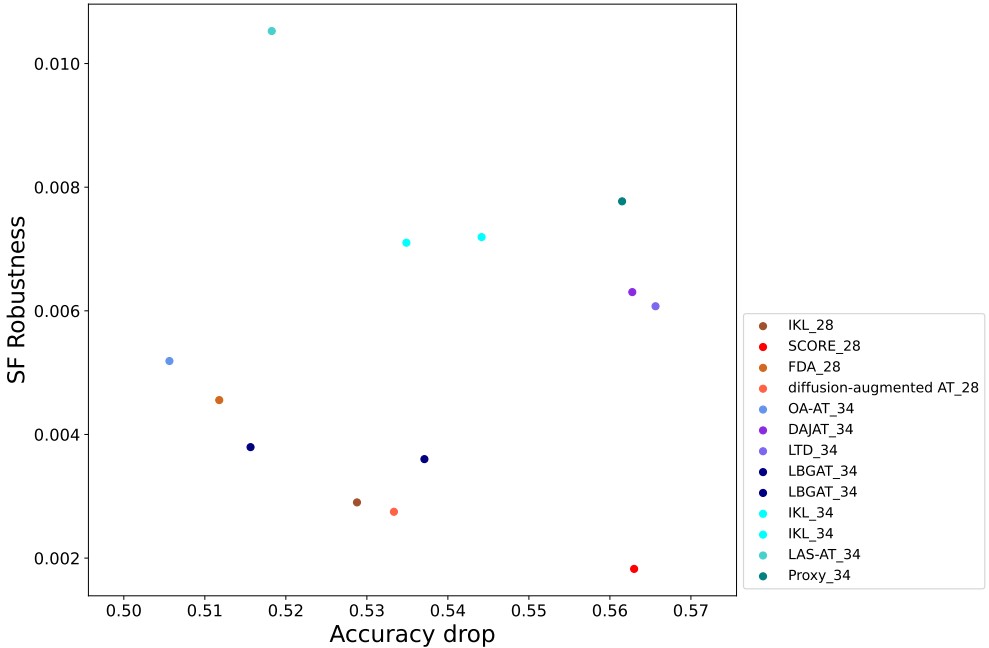

Figure 13: SF robustness and accuracy drop of various defense methods on CW in CIFAR-100. We compared two different architectures, WRN28-10 and WRN34-10, in the CIFAR-100 with 13 different defense strategies. Accuracy drop computed on CW (Carlini & Wagner, 2017). _28 and _34 refers model architecture, WRN28-10 and WRN34-10, respectively.

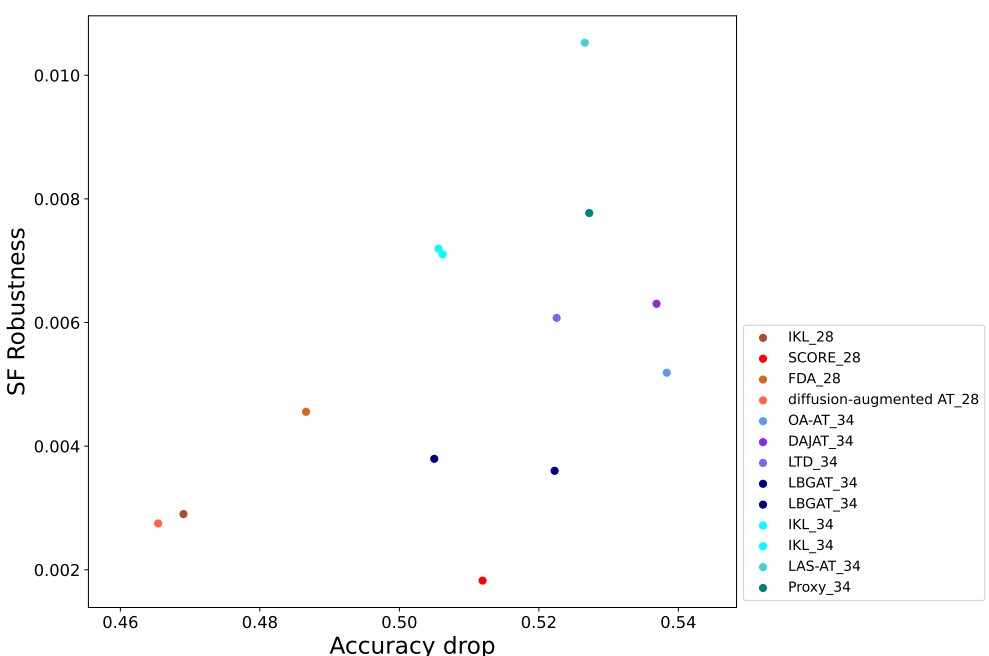

Figure 14: SF robustness and accuracy drop of various defense methods on Autoattack in CIFAR-100. We compared two different architectures, WRN28-10 and WRN34-10, in the CIFAR-100 dataset with 13 different defense strategies. Accuracy drop computed on Autoattack (Croce & Hein, 2020). _28 and _34 refers model architecture, WRN28-10 and WRN34-10, respectively.

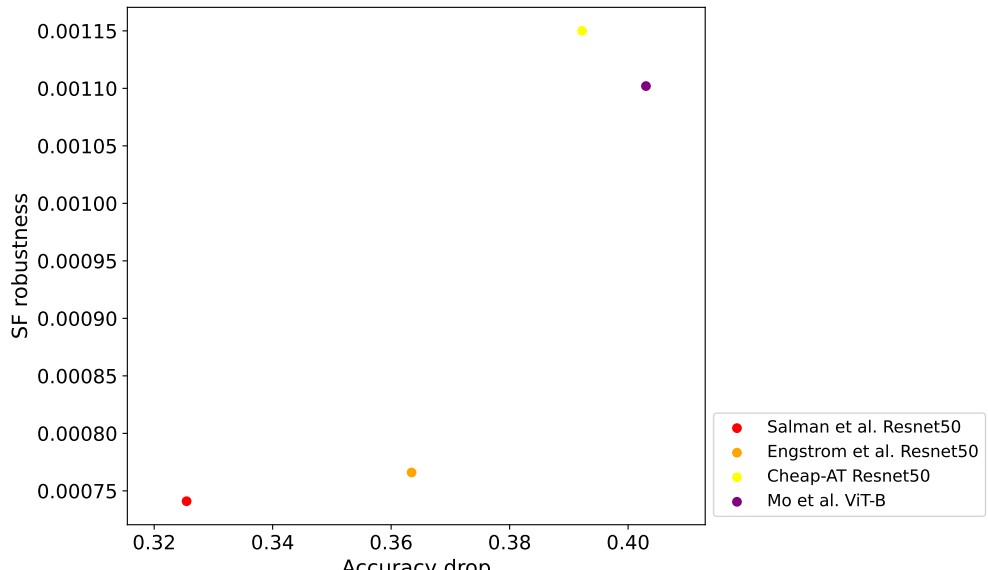

Figure 15: SF robustness and accuracy drop of various defense methods on FGSM in ImageNet. We compared two different architectures(ResNet50 and ViT-B), in ImageNet dataset, with 4 different defense strategies. Accuracy drop computed on FGSM (Goodfellow et al., 2015).

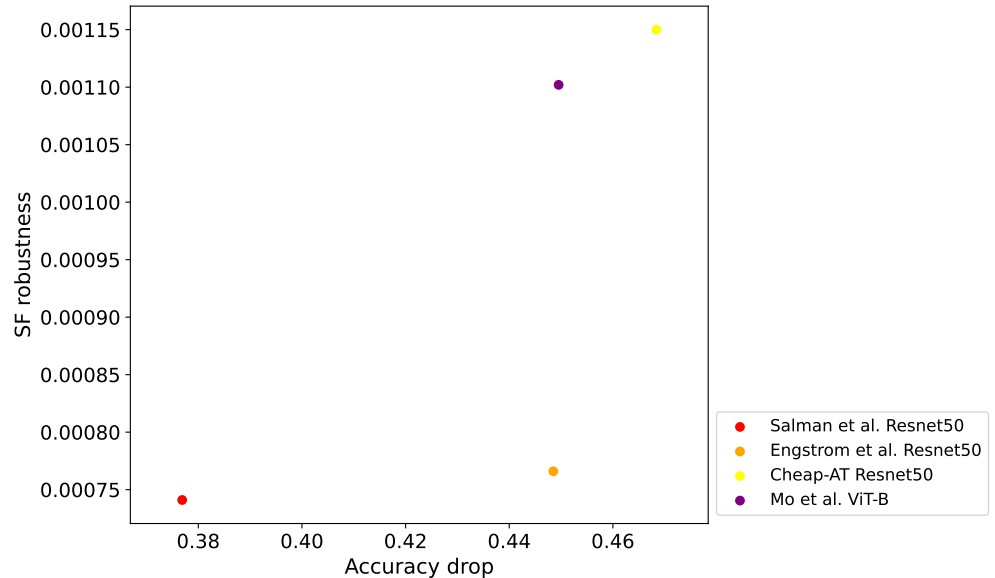

Figure 16: SF robustness and accuracy drop of various defense methods on PGD in ImageNet. We compared two different architectures(ResNet50 and ViT-B), in ImageNet dataset, with 4 different defense strategies. Accuracy drop computed on PGD (Madry, 2018).

### A.3.4 EVALUATION ON IMAGENET BENCHMARK

### A.3.5 DISCUSSION RESULTS

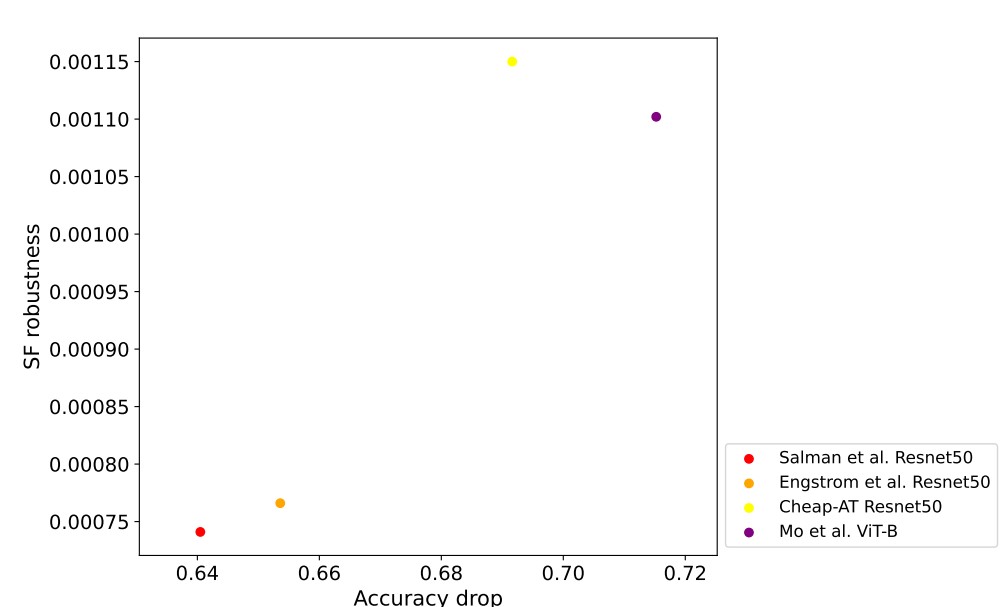

Figure 17: SF robustness and accuracy drop of various defense methods on CW in ImageNet. We compared two different architectures(ResNet50 and ViT-B), in ImageNet dataset, with 4 different defense strategies. Accuracy drop computed on CW (Carlini & Wagner, 2017).

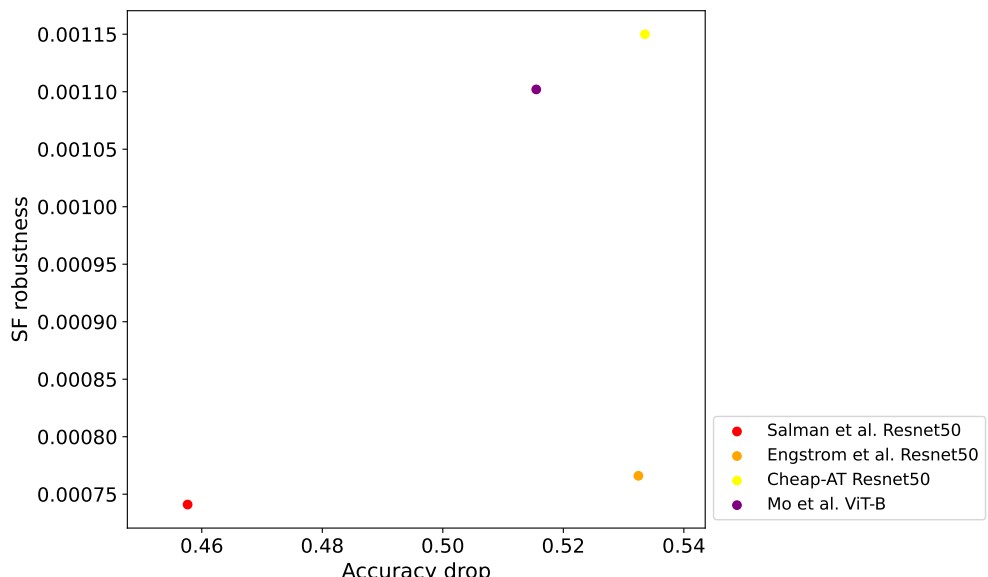

Figure 18: SF robustness and accuracy drop of various defense methods on FGSM in ImageNet. We compared two different architectures(ResNet50 and ViT-B), in ImageNet dataset, with 4 different defense strategies. Accuracy drop computed on Autoattack (Croce & Hein, 2020).

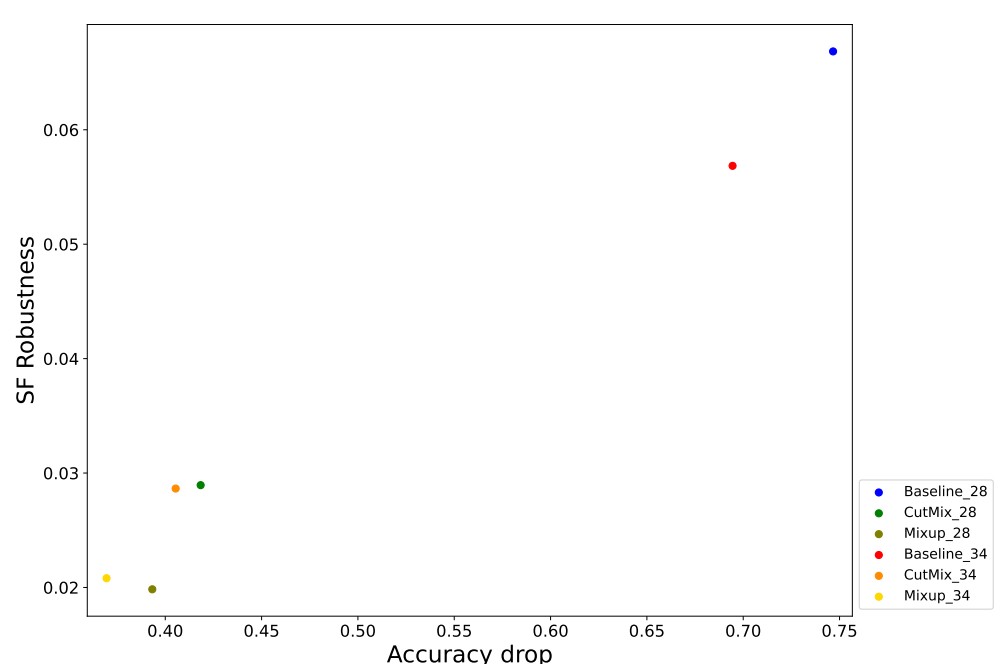

Figure 19: Comparison on models trained with various data augmentation strategies. We compared two different architectures, WRN28-10 and WRN34-10, in the CIFAR-10 dataset with three different data augmentation strategies. There is a positive correlation between SD of Shapley values and accuracy drop rate under FGSM (Goodfellow et al., 2015)

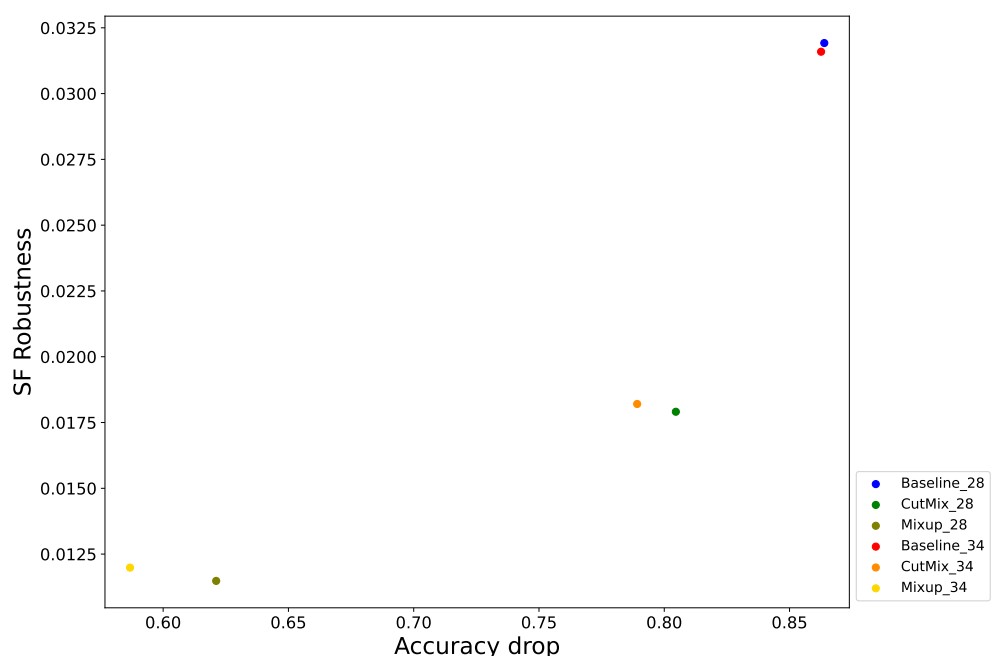

Figure 20: Comparison on models trained with various data augmentation strategies. We compared two different architectures, WRN28-10 and WRN34-10, in the CIFAR-100 dataset with three different data augmentation strategies. There is a positive correlation between SD of Shapley values and accuracy drop rate under FGSM (Goodfellow et al., 2015)

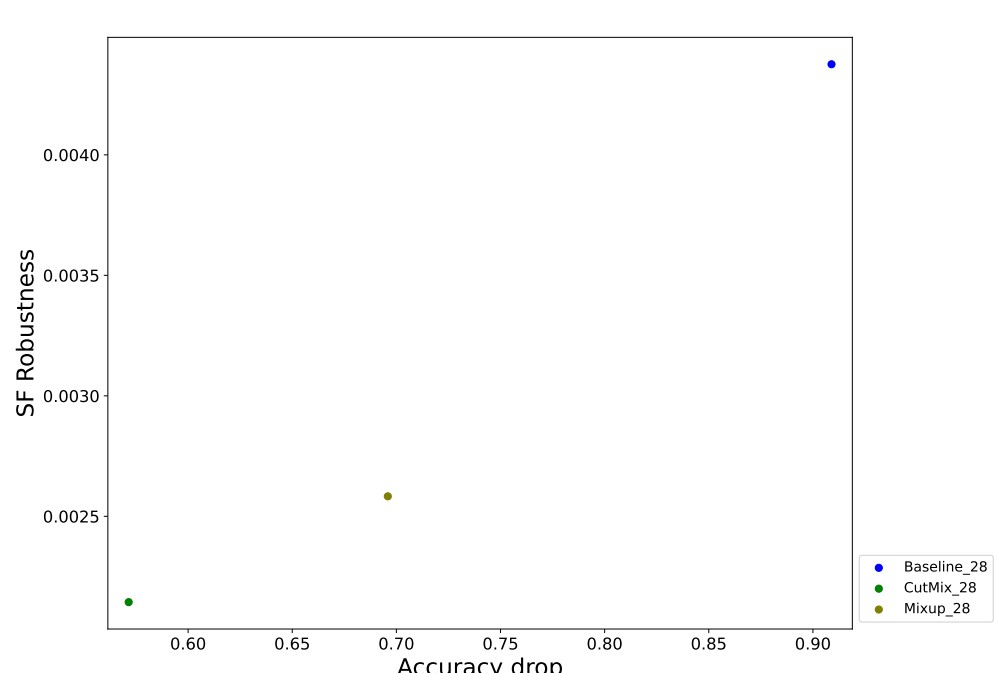

Figure 21: Comparison on models trained with various data augmentation strategies. We compared ResNet-50 in ImageNet dataset with three different data augmentation strategies. There is a positive correlation between SD of Shapley values and accuracy drop rate under FGSM (Goodfellow et al., 2015)

