# OpenReview forum: "Steady and Fair Robustness Evaluation Based on Model Interpretation"
_ICLR.cc/2025/Conference — ICLR 2025 Conference Withdrawn Submission_

### Official Review · Reviewer_Amnv · 2024-10-27

**Soundness:** 2
**Presentation:** 2
**Contribution:** 3
**Rating:** 5
**Confidence:** 4

**Summary:**

This paper highlights that traditional robustness metrics are often influenced by specific attack types, intensities, and model architectures, which may introduce biases. To address this issue, the authors introduce Shapley values and propose a Steady and Fair Robustness Evaluation Framework (SF Robustness). Experimental results demonstrate that SF Robustness can serve as a reliable tool for assessing and enhancing model robustness without extensive reliance on a variety of attack types. Moreover, this approach can be integrated with existing adversarial defense methods to improve adversarial robustness.

**Strengths:**

- This paper innovatively establishes a theoretical connection between Shapley values and adversarial robustness, providing a new perspective for robustness evaluation.
- The authors conduct experiments across multiple datasets, model architectures, and diverse adversarial training methods, demonstrating the effectiveness of the SF Robustness framework.
- Furthermore, the paper integrates Shapley values with existing adversarial training methods, achieving enhanced model robustness.

**Weaknesses:**

- The paper is largely clear and well-organized. However, certain complex mathematical derivations involving Shapley values and their Taylor approximations would benefit from further clarification (for example, how to derive Eq (4) from the definition of Shapley values), particularly for readers who may not be deeply familiar with game theory.
- Additionally, the experiments in the paper do not appear to effectively demonstrate the stability and fairness of the SF Robustness framework. While the authors showcase the correlation between SF robustness and adversarial robustness, they do not quantify the stability and fairness of SF robustness in comparison to other evaluation methods.

**Questions:**

- Could the authors provide relevant experiments to demonstrate that SF Robustness offers greater fairness and stability compared to other evaluation methods (e.g. PGD, CW, AutoAttack or their average)?
- Other problems please see Weaknesses.

---

### Official Review · Reviewer_Exzd · 2024-10-29

**Soundness:** 2
**Presentation:** 3
**Contribution:** 1
**Rating:** 1
**Confidence:** 4

**Summary:**

The work highlights the challenge of evaluating robustness in deep learning models due to the unique properties of different adversarial attacks, which complicates assessments based solely on robustness accuracy. To address this, the authors propose an alternative metric that uses the standard deviation of Shapley values to measure robustness. They demonstrate the generalizability of their method across various defense mechanisms, providing a more consistent framework for evaluating adversarial robustness.

**Strengths:**

The authors propose an alternative metric that uses the standard deviation of Shapley values to measure robustness. They demonstrate the generalizability of their method across various defense mechanisms, providing a more consistent framework for evaluating adversarial robustness.

**Weaknesses:**

1. **Motivation for this paper**: Which adversarial attacks should we trust for a fair evaluation? (lines 46-47). This is unclear. According to the definition of white-box adversarial attacks, if there is a perturbation on a given input within a specified epsilon-ball that causes the target model to make incorrect predictions, it is considered a failure. In other words, a robust model should defend against any known attack. Reliable robustness can be assessed through an ensemble of these attacks.

2. **Definition of Shapley Value**: The definition of the Shapley value is unclear. According to Equation 10, it appears that the Shapley value is derived from a Taylor approximation of specific neurons.

3. **Description of Adversarial Training**: The statement "Adversarial training aims to reduce Equation 2 during training time" (line 112) is incorrect. In fact, the objective of adversarial training is to minimize the cross-entropy or Kullback-Leibler divergence of the final predictions. While mitigating the differences in specific activations may achieve similar goals, it is not a necessary condition.

4. **Explanation of Equation 10**: The explanation of Equation 10 seems misleading (line 149). Minimizing S may drive its value close to negative infinity. I believe the author intends to express that S should approach zero.

5. **Presentation in Section 2**: The presentation in Section 2 is also misleading. The authors introduce the notations \(a^{l}\), \(w^{l}\), and \(b^{l}\) and discuss the corresponding changes caused by adversarial examples in Equation 2. Readers may conclude that each neuron's layer is relevant; however, only the penultimate layer is used.

6. **Correlation of Shapley Value and Activation Difference**: The authors indicate that the results match the correlation between the Shapley value and activation difference in Equation 10 (line 206). However, the robustness of the target model and each individual attack also have a positive correlation. I believe this result implies little and does not address the motivation presented in lines 46-47.

7. **Masked Neurons with High Shapley Value**: More specifically, can simply masking neurons with high Shapley values in a standard-trained model lead to better robust accuracy? Why do the activations across neurons become more uniform, which directly reduces the standard deviation of the neuron activations across i? (lines 212-213).

8. **Accuracy Drop Metric**: The accuracy drop used in this work is not a suitable metric. For example, Model A has a natural accuracy of 90.00% and a robust accuracy of 40% against AutoAttack; Model B has a natural accuracy of 95.00% and the same robust accuracy against AutoAttack. Obviously, Model B dominates Model A, yet it is considered the inferior model due to its accuracy drop. In an extreme case, a model with random initialization for the CIFAR100 dataset could have a natural accuracy and robust accuracy of 1%. Under the definition of accuracy drop, the corresponding value would be 0, misleading us into believing that this is a reliable model.

9. **Significance of Robust Accuracy Improvements**: The improvements in robust accuracy reported in Table 2 are not significant.

**Questions:**

1. The details of Fig 1 should be addressed, such as which dataset is used.

2. The derivation of the formula (eq 4,5,6,7,8,9,10) can be listed in the appendix.

3. The statement regarding a positive correlation is imprecise. It should provide a quantitative metric to support this claim.

4. It is difficult for readers to determine whether the data points shown in the accuracy drop vs SF robustness figures correspond to specific works.

---

### Official Review · Reviewer_Ej9o · 2024-11-05

**Soundness:** 1
**Presentation:** 2
**Contribution:** 2
**Rating:** 3
**Confidence:** 3

**Summary:**

Existing adversarial robustness evaluation methods have limitations and are easily affected by factors such as attack methods, attack intensity, and model architecture, resulting in unstable evaluation results. The author proposes a new evaluation index based on the standard deviation of the Shapley value. The core observation is that there is a correlation between the standard deviation of a neuron's Shapley value (indicating neuron importance) and the adversarial robustness of the model.

**Strengths:**

1. Proposes a adversarial robustness evaluation framework that is independent of specific attack methods
2. Provides a new research perspective by explaining adversarial robustness from the distribution of neuron importance
3. Validates the method across multiple datasets and model architectures

**Weaknesses:**

1. Although Section 2.1 provides basic theoretical analysis, it lacks rigorous proof of the necessary connection between Shapley value SD and adversarial robustness. Fails to explain why SD necessarily reflects overall model robustness
2. Correlation in Figures 3-4 is not obvious and shows many outliers. No quantitative correlation analysis (e.g., Pearson correlation coefficient). Lacks statistical significance tests for experimental results
3.Only validates four attack methods: FGSM, PGD, CW and AutoAttack. Does not consider other common attacks like DeepFool. Lacks verification against black-box attacks
4. Lacks comprehensive comparison with other robustness evaluation methods.
5. Missing analysis of cases where the evaluation method fails

**Questions:**

- Are there models with low Shapley value SD but poor adversarial robustness?
- Does the method remain effective for special architectures (VGG, DenseNet, attention mechanisms)?
- Does the evaluation framework apply to novel attack methods?
- Can SF Robustness reflect model performance under unknown attacks?
- How to determine appropriate SD thresholds for judging model robustness?

---

### Official Review · Reviewer_Vyi1 · 2024-11-08

**Soundness:** 2
**Presentation:** 2
**Contribution:** 2
**Rating:** 5
**Confidence:** 3

**Summary:**

This paper introduces Steady and Fair Robustness Evaluation, a framework to provide more stable assessments of adversarial robustness by reducing the influence of factors like attack methods and model architecture. The authors observe a strong correlation between the lower std of the Shapley values and robustness. The authors proposes extensive empirical test and also a training strategy to minimize Shapley SD as a way to enhance model robustness effectively.

**Strengths:**

1. The paper provides an alternativer perspective into studying robustness by looking at the Shapley value. The paper is overall well-written.

**Weaknesses:**

1. The paper needs an additional section about the setup and the notations. The definition of Shapley is confusing in the paper. The paper starts to discuss about the approximation of the Shapley value in section 2 even before discussing the precise definition of the Shapley value. In section 3, the paper also only gives the definition of the global Shapley value based on the approximation. Ideally, the author should propose the precise version, and then justify every step when relaxing the definition.
2. Intuitively, mitigating the Shapley value should lead to improved model robustness. The author should provide more thorough analysis on how Shapley value relate to the robustness of the a model. Under what empirical/theoretical conditions, it serves as a good metrics. Is there extreme case where a model can has small global Shapley approximation but bad robustness.
3. I think using Shapley value to study adversarial robustness is a natural idea, and I am not particular familiar with the literature here. So other reviewer can comment more on the novelty of the paper.
4. The improvement given in Table 2 seems to be small. The author should run with multiple random seeds and give a reference on the performance variance to demonstrate the significance of the improvement. The accuracy drop in Figure 5 is relatively small, and the author should provide the results of multiple runs

**Questions:**

1. Can the author give an upper bound on the Shapley value approximation? Is it possible that the approximation error get aggregated and lead to bad estimate for a deeper network?
2. Did the author consider different type of adversarial attacks? The corresponding global Shapley value approximation might be different for different types of attacks.
3. I am wondering how the author construct the adversarial example when evaluating the trained model with SF loss. Is the attacker aware of and adaptive to the additional SF loss.

---

### Note · Authors · 2024-11-13

I have read and agree with the venue's withdrawal policy on behalf of myself and my co-authors.